# Early conservation benefits of a de facto marine protected area at San Clemente Island, California

**Michael W. Esgro**[1]*, **James Lindholm**[1], **Kerry J. Nickols**[2], **Jessica Bredvik**[3]

**1** Institute for Applied Marine Ecology, California State University Monterey Bay, Seaside, CA, United States of America, **2** California State University Northridge, Northridge, CA, United States of America, **3** Naval Facilities Engineering Command Southwest, San Diego, CA, United States of America

* mesgro@gmail.com

**Data Availability Statement:** All data for this study, as well as R code used in all analyses, have been uploaded to the Harvard Dataverse public

## Abstract

De facto marine protected areas (DFMPAs) are regions of the ocean where human activity is restricted for reasons other than conservation. Although DFMPAs are widespread globally, their potential role in the protection of marine habitats, species, and ecosystems has not been well studied. In 2012 and 2013, we conducted remotely operated vehicle (ROV) surveys of marine communities at a military DFMPA closed to all civilian access since 2010 and an adjacent fished reference site at San Clemente Island, the southernmost of California's Channel Islands. We used data extracted from ROV imagery to compare density and biomass of focal species, as well as biodiversity and community composition, between the two sites. Generalized linear modeling indicated that both density and biomass of California sheephead (*Semicossyphus pulcher*) were significantly higher inside the DFMPA. Biomass of ocean whitefish (*Caulolatilus princeps*) was also significantly higher inside the DFMPA. However, species richness and Shannon-Weaver diversity were not significantly higher inside the DFMPA, and overall fish community composition did not differ significantly between sites. Demonstrable differences between the DFMPA and fished site for two highly sought-after species hint at early potential benefits of protection, though the lack of differences in the broader community suggests that a longer trajectory of recovery may be required for other species. A more comprehensive understanding of the potential conservation benefits of DFMPAs is important in the context of marine spatial planning and global marine conservation objectives.

## Introduction

Marine ecosystems worldwide are threatened by a variety of stressors, including overharvest, pollution, and climate change [1–4]. To effectively manage these complex and often interrelated problems, policymakers are increasingly adopting marine spatial planning (MSP) as a management technique [5–8]. MSP is an integrative, ecosystem-based framework that accounts for the effects of multiple human uses on marine systems and informs the spatial

repository at https://dataverse.harvard.edu/dataset.
xhtml?persistentId=doi:10.7910/DVN/XBFBBJ.

**Funding:** Funded by JL - United States Pacific Fleet
(US Army Corps of Engineers Award W9126G-12-
2-0041). https://www.usace.army.mil/. The funders
had no role in study design, data collection and
analysis, decision to publish, or preparation of the
manuscript.

**Competing interests:** The authors have declared
that no competing interests exist.

distribution of these activities. When used effectively and in concert with other marine management tools, MSP can safeguard ocean health while maintaining the delivery of essential ecosystem services [7].

Marine protected areas (MPAs), regions of the ocean set aside for conservation, are key components of MSP [6–7]. The number of MPAs has increased dramatically in recent years, with the percentage of the global ocean that is "strongly" or "fully" protected increasing from less than 0.1% to 0.6% in the last decade [9–11]. The realized ecological and economic benefits of MPAs vary widely depending on level of protection [11]. However, there is general agreement among the scientific community that strongly or fully protected MPAs can increase species density and biomass, promote the recovery of size-truncated populations, and increase biodiversity within and beyond their boundaries [9, 12].

Despite the preponderance of evidence for the conservation benefits of MPAs, very few published studies have examined the potentially similar benefits of de facto marine protected areas (DFMPAs)—places where human activity is restricted by law for reasons other than conservation or natural resource management [13]. Examples of DFMPAs include restricted areas reserved for military use [13], cable exclusion zones [14], and marine renewable energy installations such as wind turbines [15]. The first comprehensive inventory of DFMPAs in the United States indicated that there were more than 1,200 DFMPAs within U.S. waters, covering an area roughly equivalent to the total combined area protected by state and federal MPAs [16].

DFMPAs likely play a critical and heretofore unappreciated role in marine conservation [16]. On land, restricted areas such as military bases have been shown to contain higher densities of threatened and endangered species, as well as higher overall biodiversity, when compared to adjacent areas open to public access [17]. In the marine environment, Roberts et al. [18] analyzed catch data from several Florida coast fisheries and found significantly higher numbers of world-record sized catches in fisheries located near the Merritt Island National Wildlife Refuge, access to which is restricted due to the refuge's proximity to NASA's Kennedy Space Center–potential evidence of spillover from a DFMPA [18]. Rogers-Bennett et al. [19] compared historical intertidal red abalone (*Haliotis rufescens*) densities inside and outside a DFMPA (the Stornetta Ranch property) on California's central coast, which was closed to the public from 1917–2004. The authors documented 86% higher abalone densities inside the DFMPA compared to adjacent areas before the area was opened to fishing in 2004.

A more comprehensive understanding of how DFMPAs contribute to marine conservation is essential in the context of MSP. In California, for example, the Marine Life Protection Act requires that the state's system of MPAs be designed and managed as an ecologically cohesive network [20, 21]. It is likely that DFMPAs make nontrivial ecological contributions to that network, for example as sources or sinks of larval organisms, but the paucity of information regarding DFMPAs has largely precluded their incorporation into California's MPA management efforts to date.

San Clemente Island (SCI), the southernmost of the Channel Islands in the Southern California Bight (Fig 1), has been owned and managed by the United States Navy since 1934. SCI supports vital military activities that cannot be conducted anywhere else in the world [22]. At the same time, SCI's waters are home to highly productive and economically important fisheries, both commercial and recreational. Civilians also regularly use the waters surrounding SCI for non-consumptive recreational activities such as boating and scuba diving [22, 23]. However, civilian access to areas in which certain military training exercises are conducted is highly restricted and in some cases prohibited. To safely facilitate multiple human uses at SCI, the waters surrounding the island up to 3 nautical miles were divided into eight naval safety zones in June 2010 [24] (Fig 1). The type and frequency of military use, as well as associated

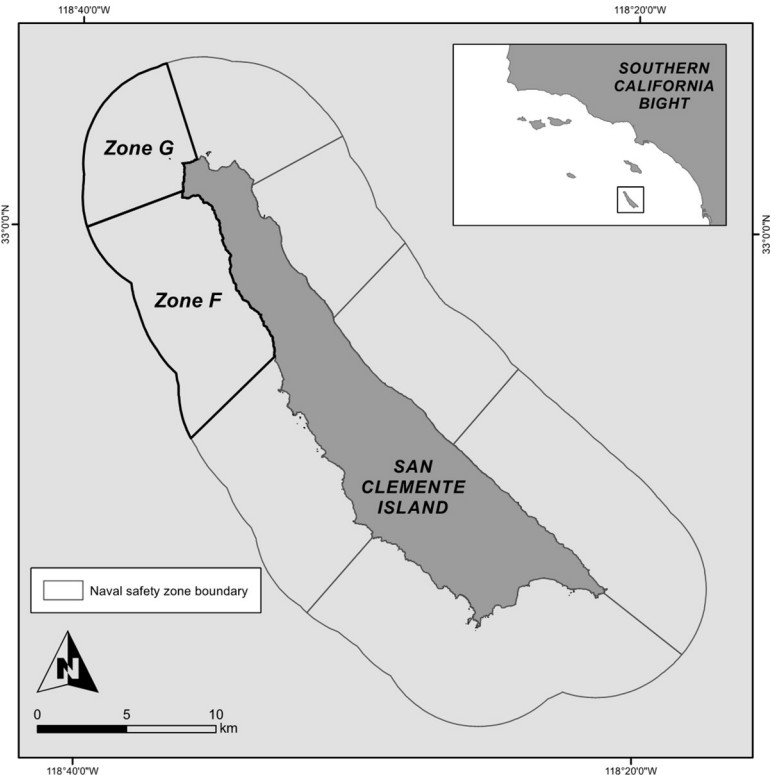

**Fig 1. San Clemente Island.** San Clemente Island is the southernmost of the Channel Islands in the Southern California Bight. To safely facilitate multiple uses at San Clemente Island, the surrounding waters up to 3 nautical miles have been divided into eight naval safety zones. This study compared marine communities at two naval safety zones at the northwest corner of the island: a DFMPA site (Zone G) and an adjacent fished reference site (Zone F).

restrictions on civilian access and activity, differ from zone to zone. Two locations, Zone G and Wilson Cove, are permanently closed to civilians. Other zones are only closed when being used for military activities that pose a threat to public safety [24]. The presence of both restricted and unrestricted areas at SCI presents a unique opportunity to compare DFMPAs to fished areas of similar habitat quality and habitat type distribution.

We used remotely operated vehicle (ROV) imagery to compare marine communities at a DFMPA site and an adjacent fished reference site at SCI. Specifically, we tested the following hypotheses: (1) density of fished species is higher at the DFMPA site than at the fished site; (2) biomass of fished species is higher at the DFMPA site than at the fished site; (3) species richness and Shannon-Weaver diversity are higher at the DFMPA site than at the fished site; and (4) fish community composition differs between sites. Our objective was to assess the potential conservation benefits of the DFMPA using these well-established MPA performance indicators.

## Materials and methods

### Study site

SCI is located 70 km west of the U.S. mainland and 30 km south of Santa Catalina Island in the Southern California Bight, and is home to a diverse assemblage of marine flora and fauna [25, 26, 27]. We compared marine communities at two sites on the northwest corner of SCI: Naval Safety Zone G (118˚38'3.259" W, 33˚2'1.831" N) and Naval Safety Zone F (118˚36'8.296" W,

32˚59'27.276" N) (Fig 1). Zone G is used regularly for Navy Sea, Air, and Land Team training, live-fire practice, and other military activity; it has been closed to all civilian access since June 2010 [24]. Zone F is open to civilians except for occasional short closures when military activities are being conducted that might threaten public safety; it is commonly fished by recreational anglers [23].

No Navy permits were required to conduct this work, as our ROV surveys fell under the purview of the San Clemente Island Integrated Natural Resources Management Plan [22] for which an Environmental Assessment and Finding of No Significant Impact was determined for natural resource management activities. We were granted explicit access to San Clemente Island by the San Clemente Island Officer in Charge.

### Image collection

ROV imagery was collected over the course of two week-long cruises in November 2012 and 2013 using a Vector M4 ROV (Deep Ocean Engineering, San Jose, California) deployed from a fishing vessel. ROV configuration and sampling protocols were based on previous studies conducted by the authors and collaborators [28, 29, 30]. The ROV was equipped with five cameras (forward-facing standard-definition video, forward-facing high-definition video, down-facing standard-definition video, digital high-definition still, and rear facing safety video), halogen lights, paired forward- and down- facing sizing lasers spaced 10 cm apart, a strobe for still photos, an altimeter, and forward-facing multibeam sonar. While at depth, the position of the ROV on the seafloor was maintained by a Trackpoint III acoustic positioning system, with the resulting coordinates logged into Hypack navigational software.

The ROV was flown over the seafloor along predetermined transect lines at a mean altitude of 1.0 m and a speed of approximately 0.67 knots. Transect placement was designed to sample a variety of depths and habitats and was based on *a priori* analysis of existing seafloor mapping data (Fig 2). While on transect, continuous video imagery was recorded from the ROV's cameras to digital tape. Still images were collected opportunistically along each transect.

### Focal species

We compared density and biomass between the two sites for focal species associated with a range of habitats and trophic levels (Table 1). We selected focal species that met the following criteria: (1) targeted for long-term monitoring in California due to ecological and/or economic importance; (2) sufficiently abundant at SCI in 2012 and 2013 to allow for reasonable sample sizes; and (3) easily identifiable in ROV video and photo imagery.

This study focused on mid-depth (40–200 m) demersal communities, as mid-depth rock habitat represents at least 75% of all marine habitats in California state waters by area and supports a high diversity of ecologically and economically important demersal fish and invertebrate species, many of which have been listed by the California Department of Fish and Wildlife as "likely to benefit" from MPA protection [31].

### Data extraction from imagery

We extracted data from forward-facing ROV video imagery by watching each transect from beginning to end and pausing video at each individual organism encountered. Video was paused to position organisms as close to the paired sizing lasers as possible. For each organism, we noted time of occurrence, count (if multiple organisms), and identification to the lowest taxonomic level possible. Identification was aided by still images and downward-facing video. Organism sizes (total lengths) were estimated to the nearest 5 cm using the paired sizing lasers and grouped into 5 cm size bins. For fishes, these lengths were later converted to weights (kg)

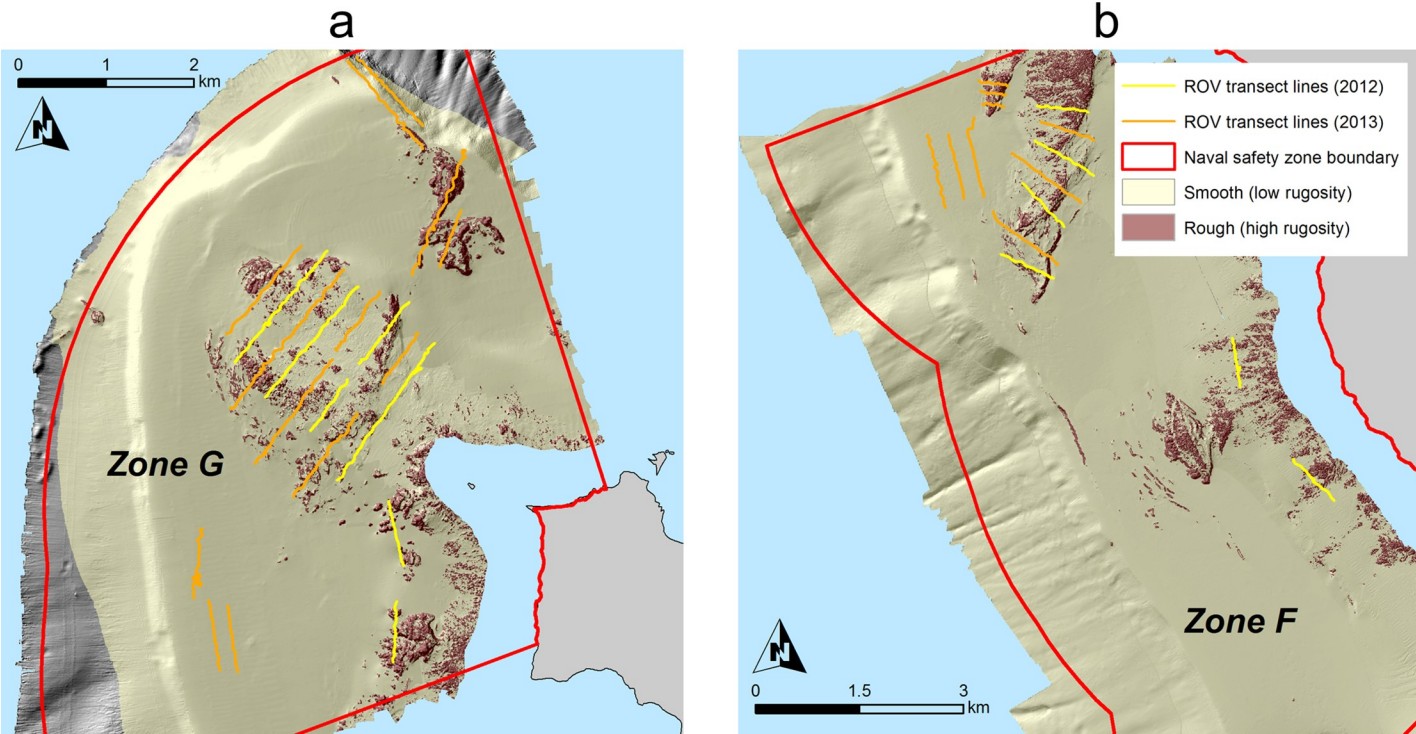

**Fig 2. Seafloor maps of a) the DFMPA site (Zone G) and b) the fished site (Zone F).** High rugosity areas indicate rocky substrate; low rugosity areas indicate sandy substrate. Transect placement was designed to encompass a variety of depths and habitat types.

using size class midpoints and the length-weight relationship

$$W = aL^b$$

where $W$ is weight of a fish in kg, $L$ is the length of that fish in cm, and $a$ and $b$ are constants unique to individual fish species (S1 Table).

Physical habitat data were collected separately by re-watching each ROV transect. Organisms were ignored during this round of data collection, and the video was paused every 1 second to record the dominant habitat ($>$ 50% of the forward-facing video frame) as rock, sand, or mixed.

## Analysis

Percent rock was calculated based on analysis of video-derived physical habitat data and represented the percentage of 1-second video frames on each transect that were classified as rock or mixed habitat. Mean transect depths were calculated from data generated by the ROV's navigational sensors, which recorded depth every second while the ROV was on transect. In general, ROV transects closely followed bathymetric contours, so depth did not vary substantially over the course of transects. Area surveyed was calculated by multiplying transect length by transect width, assumed to be 1 m for all transects based on the field of view of the ROV's cameras. In our analyses, described in more detail below, we considered these variables along with site (DFMPA or fished) as possible predictors of density, biomass, richness, and diversity, with individual transects serving as the unit of replication. Sampling year was not included as a predictor variable in our analyses, as this study was not designed for temporal comparison, i.e.

**Table 1. Focal species list and categorization.**

| Rock-associated focal species |
| :---: |
| Predatory fishes |
| Lingcod (*Ophiodon elongatus*) |
| California sheephead (*Semicossyphus pulcher*) |
| California scorpionfish (*Scorpaena guttata*) |
| Ocean whitefish (*Caulolatilus princeps*) |
| Bocaccio rockfish (*Sebastes paucispinis*) |
| Copper rockfish (*Sebastes caurinus*) |
| Olive/yellowtail rockfish (*Sebastes serranoides/S. flavidus*) |
| Vermilion/canary rockfish (*Sebastes miniatus/S. pinniger*) |
| Dwarf rockfishes |
| Dwarf-red rockfish (*Sebastes rufinanus*) |
| Halfbanded rockfish (*Sebastes semicinctus*) |
| Squarespot rockfish (*Sebastes hopkinsi*) |
| Mobile invertebrates |
| California spiny lobster (*Panulirus interruptus*) |
| Sand-associated focal species |
| Predatory fishes |
| Sanddab (*Citharichthys* spp.) |
| Surfperch (Embiotocidae, multiple species) |
| Mobile invertebrates |
| California sea cucumber (*Parastichopus californicus*) |

transects were not resampled in the second sampling year. Values for these variables associated with each transect are reported in S2 Table.

To analyze potential relatedness between predictor variables, we conducted a factor analysis of mixed data using the package FactoMineR in R [32]. Factor analysis of mixed data is designed to analyze relationships among both continuous and categorical variables; it functions as a principal components analysis for continuous data (mean depth, percent rock, area surveyed) and a multiple correspondence analysis for categorical data (site). Squared correlation coefficients determined degree of relatedness between variables. We also compared mean percent rock and mean depth between control and DFMPA transects using one-way analysis of variance (ANOVA).

For species-level comparisons, density was calculated by dividing total number of organisms on a given transect by total area surveyed during the transect. Biomass was calculated by dividing total weight of organisms on a given transect by total area surveyed during the transect. For community-level comparisons, standardized species richness was calculated by dividing the total number of unique species on a given transect by the area surveyed during the transect. Standardized species diversity was calculated by first computing the Shannon-Weaver diversity index on a given transect, and then dividing by the area surveyed during the transect.

Mean density, biomass, richness, and Shannon-Weaver diversity were assessed using the following generalized linear model with a log link function:

$$\mu \sim S + R + D$$

Given

$$Y \sim \text{QuasiPois}(\mu, \theta)$$

where *Y* is a random variable representing the ecological metric of interest, quasi-Poisson distributed with mean *μ* and variance θ, *S* is a categorical variable representing site, *R* is a continuous variable representing percent rock, and *D* is a continuous variable representing depth. We compared generalized linear models (GLMs) containing all possible combinations of predictor variables, including a null model, using Akaike's Information Criterion (AIC), AIC corrected for small sample size (AICc), and Akaike weights (AICw). We also compared means of size frequency distributions for all focal species using Kolmogorov-Smirnov tests.

To explore differences in fish community composition between sites, we calculated Bray-Curtis dissimilarity indices between all possible transect pairs. The Bray-Curtis dissimilarity index [33] quantifies the dissimilarity in species composition between two sites based on counts per area of unique fish species at each site. These calculations were based on all unique fish species observed along transects, not just focal species. Bray-Curtis dissimilarity indices were used to conduct an analysis of similarity for fish communities between sites.

All statistical analysis was conducted using R statistical software and associated packages, version 2.14.1 [34].

## Results

We conducted 15 transects (13,593.39 m$^2$ surveyed) at the fished site and 19 transects (25,264.01 m$^2$ surveyed) at the DFMPA site (Fig 2). A total of 51,688 fishes, representing 64 distinct species or species groups, and 184 mobile invertebrates, representing 8 distinct species or species groups, were observed. We also encountered a wide variety of sessile invertebrates including corals, sponges, sea whips, and sea pens in both years at both sites.

Factor analysis of mixed data indicated that site was correlated with sampling effort (i.e. more sampling occurred in the DFMPA) and percent rock was correlated with depth. However, neither percent rock nor depth was correlated with site, indicating that there were no significant differences in mean depth or mean percent rock between sites. This was confirmed by statistical comparison of mean percent rock between fished site and DFMPA site transects (one-way ANOVA, F = 0.12, p = 0.73), and mean depth between fished site and DFMPA site transects (one-way ANOVA, F = 0, p = 0.99).

Mean density and biomass for all focal species are shown in Fig 3 and reported in S2–S4 Tables. Site was found to be a significant predictor of California sheephead density, California sheephead biomass, and ocean whitefish biomass, with all of these metrics higher at the DFMPA than at the fished site. For most other focal species, percent rock and/or depth were the only significant predictors of density and biomass. For some species, no variables were found to be significant predictors of density or biomass (i.e. for these species the null model had the lowest AIC value). For density, these species were: California scorpionfish, olive/yellowtail rockfish, dwarf-red rockfish, squarespot rockfish, surfperch, and sea cucumbers. For biomass, these species were: California scorpionfish, copper rockfish, squarespot rockfish, and surfperch (Tables 2–5).

Both California sheephead and ocean whitefish showed potential filling in of size classes at the DFMPA site (Fig 4). However, means of size frequency distributions were not significantly different between sites for either species, according to Kolmogorov-Smirnov tests (p = 0.07 for sheephead, p = 0.46 for ocean whitefish).

Site and percent rock were significant predictors of species richness, with richness significantly lower at the DFMPA site, while only depth was a significant predictor of Shannon-Weaver diversity (Table 6). Non-metric multidimensional scaling based on Bray-Curtis dissimilarity indices between transects and an analysis of similarity showed no significant differences in fish communities between sites (R = 0.031, p = 0.18) (Fig 5).

**Density**

**Biomass**

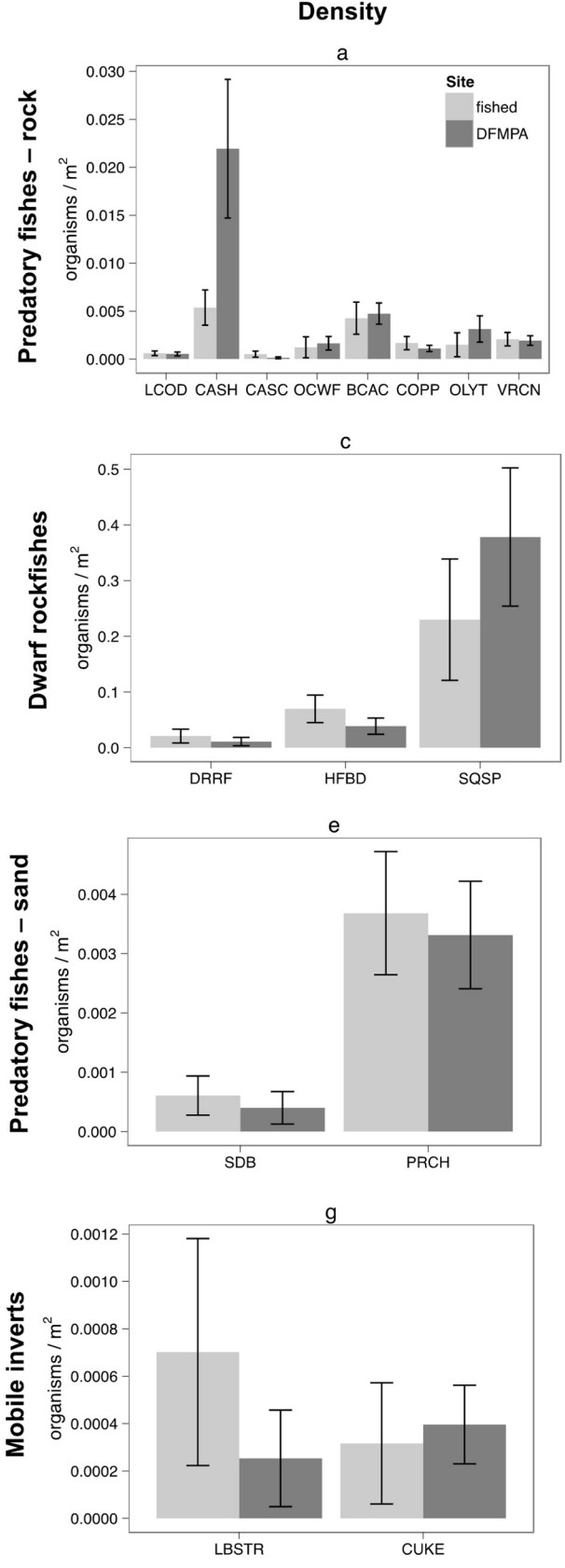

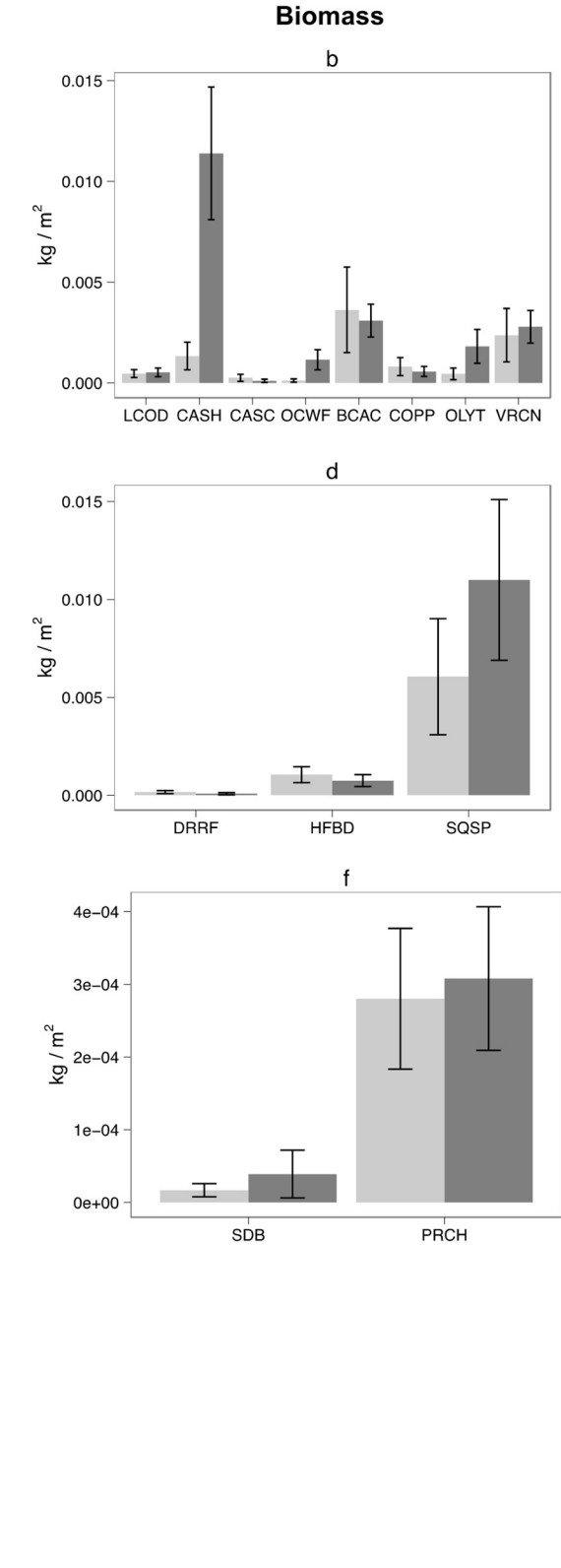

**Fig 3. Mean density and biomass for all focal species.** a) Density comparisons for rock-associated predatory fishes, b) Biomass comparisons for rock-associated predatory fishes, c) Density comparisons for dwarf rockfishes, d) Biomass comparisons for dwarf rockfishes, e) Density comparisons for sand-associated predatory fishes, f) Biomass comparisons for sand-associated predatory fishes, g) Density comparisons for mobile invertebrates. Abbreviations: LCOD = lingcod, CASH = California sheephead, CASC = California scorpionfish, OCWF = ocean whitefish, BCAC = bocaccio rockfish, COPP = copper rockfish, OLYT = olive/yellowtail rockfish, VRCN = vermilion/canary rockfish, DRRF = dwarf-red rockfish, HFBD = halfbanded rockfish, SQSP = squarespot rockfish, SDB = sanddab, PRCH = perch, LBSTR = California spiny lobster, CUKE = California sea cucumber.

## Discussion

Demonstrable differences between the DFMPA and fished sites for two highly sought-after fishes, California sheephead and ocean whitefish, hint at early potential benefits of protection, though the lack of differences in the broader community suggests a longer trajectory of recovery may be required for other species. We hypothesized that reduced fishing pressure in the DFMPA might result in conservation benefits similar to those that have been documented in MPAs across the globe—namely, increased density and biomass of some fished species, as well

**Table 2. GLM results for rock-associated focal species density.** Models shown are those with the lowest AICc of all candidate models considered.

| | Variable | Coefficient | p-value | df | AIC | AICc | AICw |
|---|---|---|---|---|---|---|---|
| **Rock-associated focal species** | | | | | | | |
| Predatory fishes | | | | | | | |
| Lingcod | | | | 3 | -381.03 | -380.25 | 0.51 |
| | Percent rock | 1.36E-05 | 1.86E-02 | | | | |
| California sheephead | | | | 5 | -164.40 | -162.25 | 0.50 |
| | Site | 1.58E-02 | 2.86E-02 | | | | |
| | Percent rock | 2.58E-04 | 6.47E-02 | | | | |
| | Depth | -3.19E-04 | 5.63E-03 | | | | |
| California scorpionfish | | | | 2 | -379.20 | -378.81 | 0.34 |
| | NA | | | | | | |
| Ocean whitefish | | | | 3 | -286.08 | -285.28 | 0.38 |
| | Depth | -3.66E-05 | 4.53E-02 | | | | |
| Bocaccio rockfish | | | | 3 | -258.65 | -257.85 | 0.46 |
| | Percent rock | 8.67E-05 | 1.33E-02 | | | | |
| Copper rockfish | | | | 3 | -322.70 | -321.90 | 0.23 |
| | Percent rock | 2.28E-05 | 8.64E-02 | | | | |
| Olive/yellowtail rockfish | | | | 2 | -254.65 | -254.27 | 0.26 |
| | NA | | | | | | |
| Vermilion/canary Rockfish | | | | 4 | -315.89 | -314.51 | 0.47 |
| | Percent rock | 4.17E-05 | 7.87E-03 | | | | |
| | Depth | 2.33E-05 | 5.49E-02 | | | | |
| Dwarf rockfishes | | | | | | | |
| Dwarf-red rockfish | | | | 2 | -119.73 | -119.34 | 0.31 |
| | NA | | | | | | |
| Halfbanded rockfish | | | | 3 | -76.12 | -75.32 | 0.27 |
| | Depth | 8.99E-04 | 2.62E-02 | | | | |
| Squarespot rockfish | | | | 2 | 51.09 | 51.47 | 0.38 |
| | NA | | | | | | |
| Mobile invertebrates | | | | | | | |
| California spiny lobster | | | | 3 | -348.90 | -348.10 | 0.32 |
| | Depth | -1.24E-05 | 8.48E-02 | | | | |

**Table 3. GLM results for sand-associated focal species density.** Models shown are those with the lowest AICc of all candidate models considered.

| | Variable | Coefficient | p-value | df | AIC | AICc | AICw |
|---|---|---|---|---|---|---|---|
| **Sand-associated focal species** | | | | | | | |
| Predatory fishes | | | | | | | |
| Sanddab | | | | 3 | -380.15 | -379.35 | 0.50 |
| | Percent rock | -3.31E-05 | 1.20E-06 | | | | |
| Surfperch | | | | 2 | -277.30 | -276.91 | 0.40 |
| | NA | | | | | | |
| Mobile invertebrates | | | | | | | |
| California sea cucumber | | | | 2 | -382.22 | -381.84 | 0.38 |
| | NA | | | | | | |

as community-wide effects including increased species richness and diversity [9]. However, we recognized that the relatively short two to three-year period of "recovery" at the time of the ROV surveys may limit those observed benefits [35–36]. Indeed, we observed very limited

**Table 4. GLM results for rock-associated focal species biomass.** Models shown are those with the lowest AICc of all candidate models considered.

| | Variable | Coefficient | p-value | df | AIC | AICc | AICw |
|---|---|---|---|---|---|---|---|
| **Rock-associated focal species** | | | | | | | |
| Predatory fishes | | | | | | | |
| Lingcod | | | | 3 | -384.10 | -383.32 | 0.44 |
| | Percent rock | 1.16E-05 | 3.41E-02 | | | | |
| California sheephead | | | | 5 | -216.69 | -214.55 | 0.57 |
| | Site | 9.69E-03 | 4.77E-03 | | | | |
| | Percent rock | 1.19E-04 | 6.64E-02 | | | | |
| | Depth | -1.34E-04 | 1.13E-02 | | | | |
| California scorpionfish | | | | 2 | -414.98 | -414.60 | 0.32 |
| | NA | | | | | | |
| Ocean whitefish | | | | 4 | -336.26 | -334.88 | 0.19 |
| | Site | 1.03E-03 | 7.18E-02 | | | | |
| | Depth | -1.35E-05 | 1.14E-01 | | | | |
| Bocaccio rockfish | | | | 3 | -252.79 | -251.99 | 0.47 |
| | Percent rock | 8.97E-05 | 1.82E-02 | | | | |
| Copper rockfish | | | | 2 | -347.48 | -347.09 | 0.33 |
| | NA | | | | | | |
| Olive/yellowtail rockfish | | | | 3 | -299.44 | -298.64 | 0.23 |
| | Percent rock | 3.12E-05 | 9.56E-02 | | | | |
| Vermilion/canary rockfish | | | | 4 | -280.73 | -279.35 | 0.72 |
| | Percent rock | 7.52E-05 | 4.57E-03 | | | | |
| | Depth | 5.87E-05 | 5.32E-03 | | | | |
| Dwarf rockfishes | | | | | | | |
| Dwarf-red rockfish | | | | 3 | -460.71 | -459.91 | 0.26 |
| | Percent rock | 2.67E-06 | 1.25E-01 | | | | |
| Halfbanded rockfish | | | | 4 | -349.38 | -348.00 | 0.32 |
| | Percent rock | 1.51E-05 | 1.03E-01 | | | | |
| | Depth | 1.77E-05 | 1.85E-02 | | | | |
| Squarespot rockfish | | | | 2 | -184.36 | -183.98 | 0.38 |
| | NA | | | | | | |

**Table 5. GLM results for sand-associated focal species biomass.** Models shown are those with the lowest AICc of all candidate models considered.

| | Variable | Coefficient | p-value | df | AIC | AICc | AICw |
|---|---|---|---|---|---|---|---|
| **Sand-associated focal species** | | | | | | | |
| Predatory fishes | | | | | | | |
| Sanddab | | | | 3 | -527.14 | -526.34 | 0.50 |
| | Percent rock | -1.87E-06 | 6.25E-03 | | | | |
| Surfperch | | | | 2 | -432.38 | -431.99 | 0.35 |
| | NA | | | | | | |

evidence for our hypotheses, with only 1 of 15 focal species showing increases in density and 2 of 15 focal species showing increases in biomass at the DFMPA site.

California sheephead exhibited the most striking result, a ten-fold increase in both density and biomass at the DFMPA site. California sheephead are highly sought after by recreational anglers in Southern California (240,305 individuals taken in state waters from 2010–2018 [37]); level of historical fishing pressure is one of the most important drivers of the rate of population recovery inside MPAs [36, 38]. Large male sheephead in particular are preferred targets for fishermen [39–40]. Since large males monopolize access to female sheephead, their removal can dramatically reduce the species' overall reproductive rate in fished areas [39, 41]. This effect is compounded by the fact that sheephead are protogynous sequential hermaphrodites, which means that in the absence of a male reproduction is halted until a female can transition sexes and take its place [39]. However, sheephead mature and reproduce relatively quickly, especially compared to slower-growing species such as rockfish [42], meaning that populations may be able to recover on a relatively short time scale following the removal or reduction of fishing pressure. These unique life history characteristics, coupled with high historical fishing pressure, make sheephead a potential bellwether for ecological changes resulting from protection.

Biomass of ocean whitefish was significantly higher at the DFMPA site. Like sheephead, ocean whitefish are commonly fished by recreational anglers in Southern California (683,338 individuals taken in state waters from 2010–2018 [37]) and mature relatively quickly (sexually mature at 3–5 years) [43]. Increases in mean body size and filling in of size-truncated populations is a well-documented response for species likely to benefit from protection [9, 35–36]. Indeed, both sheephead and ocean whitefish showed potential filling in of size classes at the DFMPA site (Fig 4), but means of size frequency distributions were not found to be significantly different between sites for either species. It is possible that larger size classes of sheephead and whitefish will continue to fill in at the DFMPA site with increased recovery time. We did not find a positive relationship between protection and biomass, or evidence of changes in size frequency distributions, for any of the other focal species considered.

Lack of observable differences between sites may have been due, at least in part, to environmental differences, spillover and site selection, or other human uses. Habitat/microhabitat type, quality, and availability are critical drivers of marine species distribution and community composition, and in some cases are more influential than the presence or absence of protection from fishing [44–47]. In addition, physical and chemical oceanographic conditions can have significant impacts on marine communities, for example by driving patterns of larval dispersal or influencing nutrient availability in an ecosystem [48–50]. These factors have the potential to override or confound any potential benefits of removing or reducing fishing pressure. For example, the size frequency distribution for sheephead shows that not only are larger fish present in the DFMPA, but also new recruits and small juvenile size classes are present as well (Fig 4). This could indicate better habitat quality for all size classes of this species in the DFMPA.

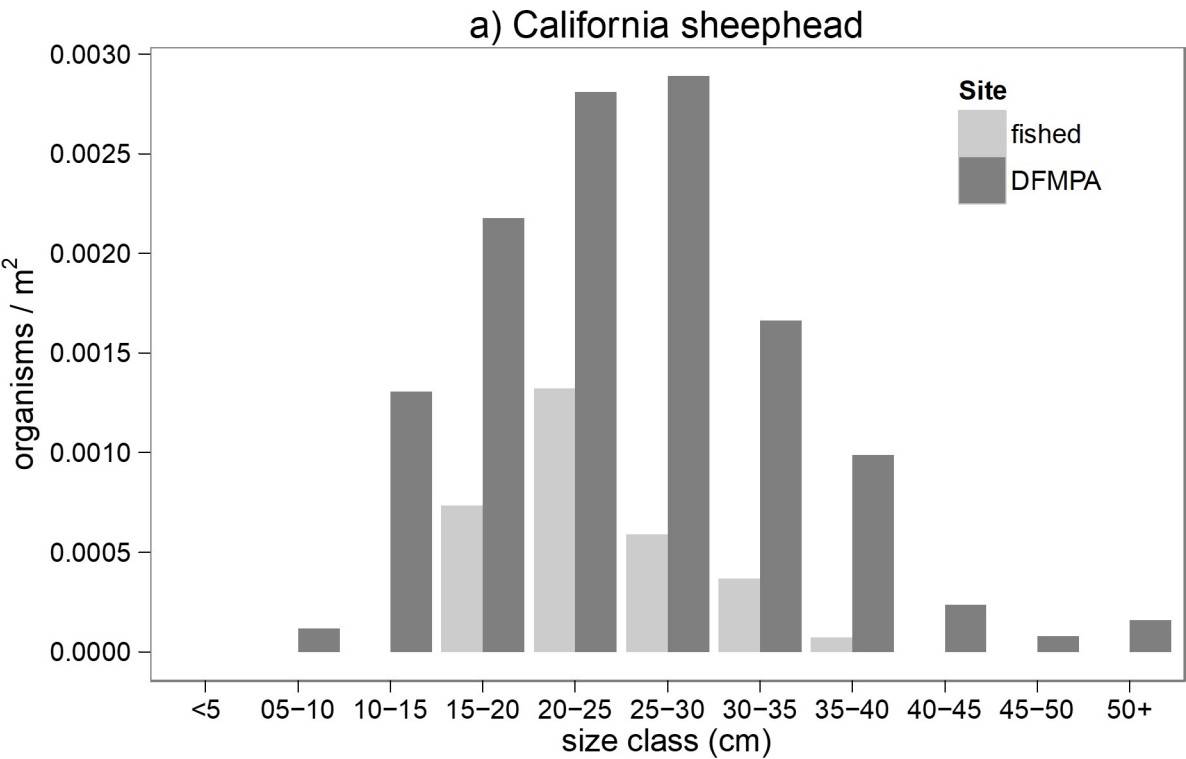

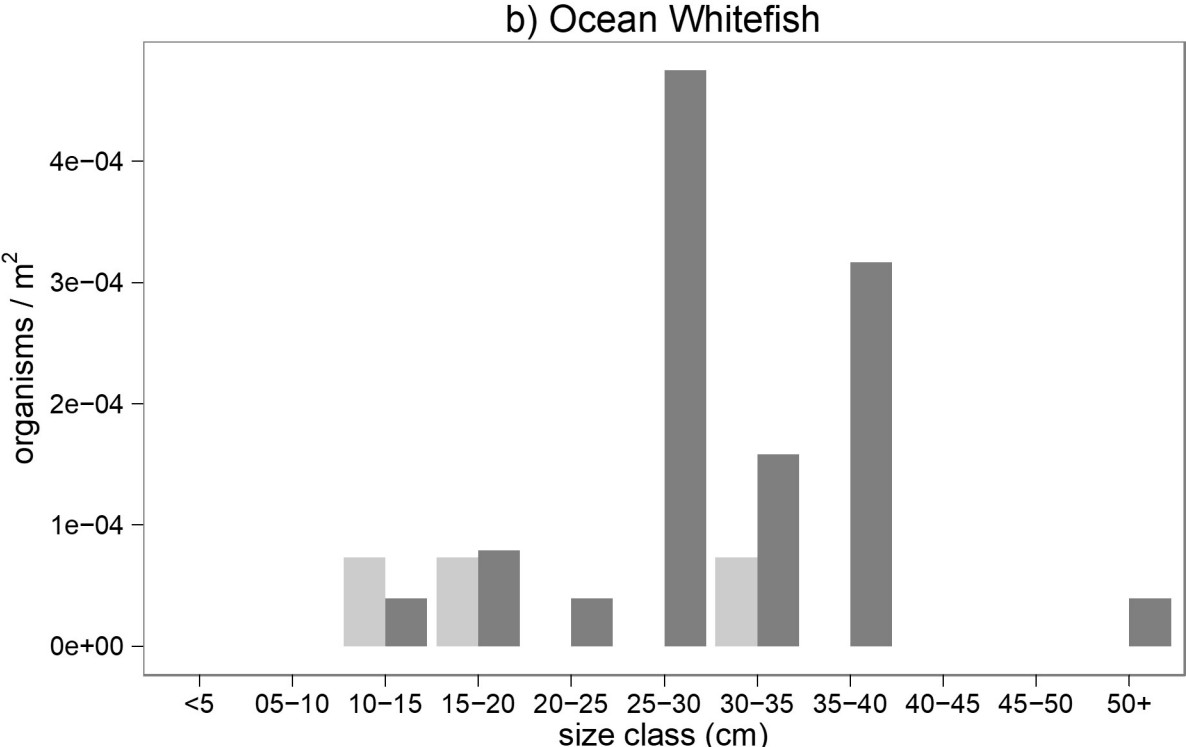

**Fig 4. Size frequency distribution comparisons.** Comparisons of a) California sheephead and b) Ocean whitefish size frequency distributions at the DFMPA and fished sites. Both species showed potential filling in of size classes at the DFMPA site, but means of the distributions were not significantly different between sites for either species.

**Table 6. GLM results for mean standardized species richness and mean standardized Shannon-Weaver diversity.** Models shown are those with the lowest AICc of all candidate models considered.

| | Variable | Coefficient | p-value | df | AIC | AICc | AICw |
|---|---|---|---|---|---|---|---|
| Richness | | | | 4 | -234.69 | -233.44 | 0.63 |
| | Site | -6.20E-03 | 1.82E-02 | | | | |
| | Percent rock | 2.00E-04 | 1.00E-04 | | | | |
| Diversity | | | | 3 | -382.10 | -381.30 | 0.24 |
| | Depth | -6.68E-06 | 1.29E-01 | | | | |

Comparisons of mean depth and percent rock between sites analyses indicated that the DFMPA and fished sites were similar in terms of habitat, an important consideration given the fact that depth and percent rock were found to be significant predictors of many of the ecological metrics we examined. However, further study using advanced habitat suitability modeling techniques would allow for a more fine-scale comparison of the distribution of suitable habitat for focal species between sites [30, 51]. In particular, it would be important to compare the rugosity and complexity of rocky habitat, rather than just percent rock, across sites.

A lack of ecological divergence between sites may also have been a result of the spillover effect. Spillover of adult or larval organisms from MPAs to unprotected waters is widely acknowledged as an economic benefit of spatial protection [18, 52–53]. However, spillover may confound spatial comparison if organisms are exported from a protected site to a reference site. This confounding factor is especially important to consider when the protected and

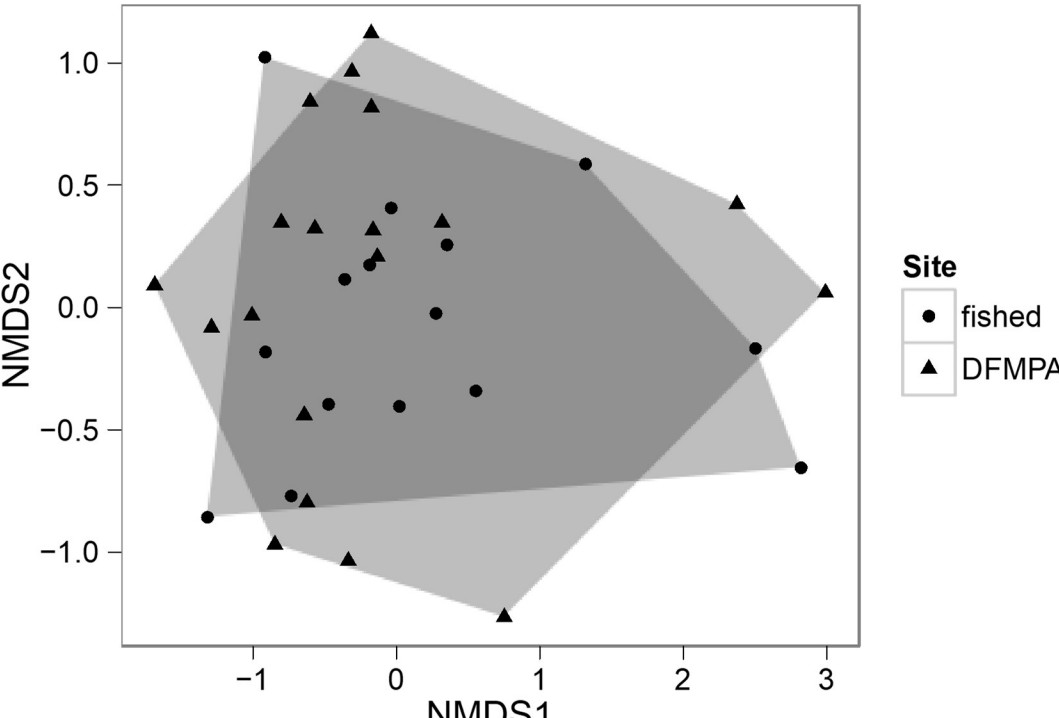

**Fig 5. Non-metric multidimensional scaling plot plot of fish community composition at DFMPA and fished sites.** Non-metric multidimensional scaling plot based on Bray-Curtis dissimilarity indices between fished and DFMPA transects. Transects are not clustered according to site, suggesting that fish community composition was not significantly different between sites.

control sites are close together [54], as was the case for the sites in this study. Future studies at SCI could overcome this limitation by surveying multiple DFMPA and reference sites, as discussed in more detail below, and could also consider the possibility of differential spillover based on the reproductive differences between various species (e.g. variation in pelagic larval durations).

Site selection may have influenced our results in ways beyond the potential confounding effects of adult and larval spillover. The Navy does not collect data on recreational fishing activities, so our understanding of the spatial distribution of fishing effort at SCI, as well as historical mortality rates for fished species, is limited. However, it is reasonable to assume that recreational anglers might limit fishing in Zone F, as it is directly adjacent to the DFMPA and anglers may be wary of accidentally entering a restricted area. Furthermore, Zone F is on the windward side of the island and far from the most popular anchorage, so fishing pressure may be limited here compared to other locations on the island. If fishing is indeed limited in Zone F compared to other open areas at SCI, mortality may be spread across many species, dampening any potential recovery signal especially in a short time frame. Finally, Zone G is located at the northern tip of the island. Many studies show higher densities and biomass of fishes at the tips of islands, reefs, and atolls, due to stronger currents, higher productivity, and increased prey availability in these areas [55]. To address these potential confounding factors, future work at SCI could include multiple DFMPA and reference sites in an expanded study design. Zone W (Wilson Cove) and Zone D (the Shore Bombardment Area) are both heavily restricted and are therefore ideal candidate DFMPAs that could be incorporated into a larger-scale analysis. Reference sites for these additional DFMPAs could then be selected so that sites being compared are separated by multiple units of dispersal for species of interest [54].

When considering potential conservation benefits of DFMPAs, it is essential to consider other human uses, most importantly the underlying reason for DFMPA establishment. Unlike MPAs, DFMPAs are generally not managed to achieve conservation goals. Therefore, DFMPAs may have neutral or even negative effects on marine communities, depending on the type and amount of human activity conducted within their boundaries. However, it is unlikely that military activity conducted inside this particular DFMPA has directly adverse effects on marine life. Environmental impact studies conducted at SCI have consistently found that the Navy's training and testing activities have negligible impact on marine species and habitats [56–57]. Moreover, due to the fact that fishing places such substantial pressure on marine ecosystems, any effects of military activity inside the DFMPA are likely to be substantially less important from a conservation perspective than the associated reduction of fishing pressure.

The potential conservation benefits of DFMPAs are likely to become increasingly important in the context of current global objectives for marine protection. In 2010, the United Nations Convention on Biological Diversity adopted the Aichi Biodiversity Targets to "safeguard ecosystems, species, and genetic diversity," among other goals [58]. Aichi Target 11 calls for the protection of at least 17% of terrestrial and inland water areas, and 10% of coastal and marine ecosystems, by 2020. The International Union for the Conservation of Nature is now advocating for an even more aggressive goal–"30 by 30," or 30% of the world's ocean protected by 2030 [59]. With only 3.5% of the global ocean currently covered by MPAs, formal protection would have to steeply increase for these targets to be met. However, both Target 11 and the International Union for Conservation of Nature's 30 by 30 goal include "other effective area-based conservation measures" (OECMs) as potential alternatives to formal MPAs for marine protection. Many DFMPAs could potentially be considered OECMs.

The definition of an OECM, and how such areas may contribute to biodiversity conservation, have been the subject of much debate. In particular, the Convention on Biological Diversity has faced pressure to keep the definition as broad as possible so that parties to the

Convention can claim to be keeping to their commitments and making progress toward Target 11 [60]. In 2018, the CBD adopted the following definition of an OECM:

*"A geographically defined area other than a Protected Area, which is governed and managed in ways that achieve positive and sustained long-term outcomes for the in situ conservation of biodiversity with associated ecosystem functions and services and where applicable, cultural, spiritual, socio–economic, and other locally relevant values."* [61].

Parties to the Convention will look to this definition as they begin to implement a post-2020 biodiversity framework this year [62]. This means that several unanswered scientific questions surrounding OECMs will need to be addressed. First, OECMs will need to be monitored and compared to unprotected reference sites to ensure that they are indeed achieving biodiversity conservation goals. Second, the governance structures of OECMs will need to be studied and assessed, with particular attention paid to issues of equity. For example, military restricted areas may provide significant ecological benefits, but may not contribute to the conservation of cultural or spiritual values. Finally, if "effective conservation" is interpreted to apply to all species rather than just a select few, the conservation benefits of OECMs will need to be assessed at the ecosystem level. For example, while the DFMPA examined in this study does demonstrate tangible conservation benefits for two heavily fished species, we did not directly assess how routine military operations or periodic exercises affect all species in the area, including marine mammals, beyond a review of the Navy's EIS for SCI [56–57]. Such considerations are complex and, although relatively well explored in the MPA literature, remain largely unanswered for OECMs.

This study, along with prior work in this area, suggests that DFMPAs are likely to add value to existing MPA systems. We suggest that agencies and entities involved with large-scale marine management and conservation more explicitly consider DFMPAs in their decision-making and explore the possibility of working with scientific, conservation, and indigenous communities to achieve both the primary management goal of DFMPAs (e.g. military priorities) as well as biodiversity conservation. However, the effective integration of DFMPAs into marine management requires an improved understanding of the contributions that these unique areas can make to global and regional conservation objectives. As demonstrated here, this knowledge gap can be addressed through long-term, robust biological and environmental monitoring inside DFMPAs and at unprotected reference sites. We suggest continued, community-wide ecological assessments of DFMPAs as well as more research into the contributions these areas may make to social, economic, cultural, and spiritual values.

## Conclusion

To our knowledge, this study is the first spatially explicit, community-wide comparison of marine ecosystems inside and outside a DFMPA. It provides evidence that DFMPAs may provide conservation benefits similar to those of MPAs. Our results encourage further exploration of the role that DFMPAs may play in marine conservation, and especially their potential integration into existing MSP frameworks and plans to achieve global conservation goals.

## Supporting information

**S1 Table. Length-weight relationship parameter values and sources for focal fish species.**
(DOCX)

**S2 Table. Variables (name, year, site, percent rock, mean depth, and area surveyed) associated with ROV transects.**
(DOCX)

**S3 Table. Means and standard errors for focal species density at fished and DFMPA sites.** (DOCX)

**S4 Table. Means and standard errors for focal species biomass at fished and DFMPA sites.** (DOCX)

## Acknowledgments

Generous financial support for this project came from the United States Pacific Fleet (US Army Corps of Engineers Award W9126G-12-2-0041) and private donations. We thank the crews of the F/V *Donna Kathleen* and Marine Applied Research and Exploration for key assistance in the field. Finally, we thank two anonymous reviewers, whose suggestions and contributions greatly improved this manuscript.

## Author Contributions

**Conceptualization:** Michael W. Esgro, James Lindholm.

**Data curation:** Michael W. Esgro, James Lindholm.

**Formal analysis:** Michael W. Esgro, James Lindholm, Kerry J. Nickols.

**Funding acquisition:** James Lindholm.

**Investigation:** Michael W. Esgro, James Lindholm.

**Methodology:** Michael W. Esgro, James Lindholm, Kerry J. Nickols.

**Project administration:** James Lindholm.

**Resources:** James Lindholm.

**Software:** Michael W. Esgro.

**Supervision:** Michael W. Esgro, James Lindholm, Kerry J. Nickols.

**Validation:** Michael W. Esgro, James Lindholm.

**Visualization:** Michael W. Esgro, Kerry J. Nickols.

**Writing – original draft:** Michael W. Esgro.

**Writing – review & editing:** Michael W. Esgro, James Lindholm, Kerry J. Nickols, Jessica Bredvik.

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
