## [Decision Letter · Decision Letter 0]

15 Nov 2019

PONE-D-19-27751

Early conservation benefits of a de facto marine protected area at San Clemente Island, California

PLOS ONE

Dear Mr. Esgro,

Thank you for submitting your manuscript to PLOS ONE. After careful consideration, we feel that it has merit but does not fully meet PLOS ONE’s publication criteria as it currently stands. Therefore, we invite you to submit a revised version of the manuscript that addresses the points raised during the review process.

I found this to be a really interesting and generally well written manuscript and both reviewers agreed. However, both reviewers have also identified some areas where the manuscript could be improved, mainly around clarifying points and methods in the manuscript. I have also provided some editorial comments. Overall, I think these comments are relatively minor and can be easily addressed. I encourage the authors to address all the comments provided when making their revisions.

We would appreciate receiving your revised manuscript by Dec 30 2019 11:59PM. To enhance the reproducibility of your results, we recommend that if applicable you deposit your laboratory protocols in protocols.io, where a protocol can be assigned its own identifier (DOI) such that it can be cited independently in the future. For instructions see: http://journals.plos.org/plosone/s/submission-guidelines#loc-laboratory-protocols

We look forward to receiving your revised manuscript.

Kind regards,

Heather M. Patterson, Ph.D.

Academic Editor

PLOS ONE

Journal Requirements:

Reviewers' comments:

Reviewer's Responses to Questions

**Comments to the Author**

1. Is the manuscript technically sound, and do the data support the conclusions?

Reviewer #1: Yes

Reviewer #2: Partly

2. Has the statistical analysis been performed appropriately and rigorously? 

Reviewer #1: Yes

Reviewer #2: Yes

3. Have the authors made all data underlying the findings in their manuscript fully available?

Reviewer #1: Yes

Reviewer #2: No

4. Is the manuscript presented in an intelligible fashion and written in standard English?

Reviewer #1: Yes

Reviewer #2: Yes

5. Review Comments to the Author

Reviewer #1: This is a clear and compelling paper that evaluates some of the effects of restricted use in a de facto marine protected area (DFMPA) off the California coast. The subject is of particular interest as countries look for ways to ramp up the percent protected area coverage of their maritime jurisdictions. Military restrictions around San Clemente Island present an opportunity to study a controlled area and compare biomass and community structure with adjacent fished areas. The paper is technically sound, extremely well-written, and presents the data that were used in the analysis.

However, regarding whether the data can be said to support the conclusions, this is the paper’s greatest weakness. The authors claim that the lack of differentiation in biodiversity metrics between the DFMPA and fished sites for most species surveyed is due to longer trajectories of recovery for species other than sheephead and ocean whitefish – but does what we know about the life histories the other species support this contention? An alternative hypothesis might be that fishing pressure in the fished area is spread across many species, and therefore does not alter the community ecology (therefore not presenting a condition from which recovery can be measured). Or perhaps microhabitats do vary between fished and DFMPA sites – these were not assessed and as the authors themselves state, may be a bigger factor in determining condition (biomass and diversity) than fishing pressure. Differential spillover is unlikely, unless the authors can point to some aspect of reproductive ecology or larval dispersion of the sheephead and whitefish not shared by the other species. Since the data collected and the time since closure are both limited, it may be too early and too much of a stretch to speculate as to why the DFMPA has bigger specimens and higher densities of sheephead and whitefish and not the other species.

I believe the paper would also benefit from an expanded discussion around Other Effective Conservation Measures (OECMs). The authors are undoubtedly aware of the very fluid (and sometimes highly contentious) debates around what constitutes OECMs, and the fact that there is a lot of pressure to make the definition as wide as possible so that contracting parties to the Convention on Biological Diversity can claim to be approaching Aichi Target 11, thereby keeping to their commitments. However, the draft decision of the SBBSTA (CBD/SBSTTA/22/L.2 6 July 2018) does say that the definition of OECM covers considerations of governance, including equity issues, which military bases and restricted areas could not be said to fulfill. Furthermore, while the DFMPA studied does demonstrate tangible conservation benefits for two heavily fished species, no assessment was made of how routine military operations or periodic exercises affect all species in the area, including marine mammals. ‘Effective conservation’ within the OECM means conservation of all species, not only a select few. The paper should be forthright about mentioning these limitations.

Another area where the paper could be strengthened is in the discussion of future research – in particular how to overcome the design flaw that the current study has in that it had no reference points in fished areas other than those adjacent to the DFMPA. Spillover could confound the existing comparison, as could any perception issues among fishers that would limit their fishing in the areas adjacent to the restricted area (if, for instance, fishers were wary of accidentally entering the restricted area). If any information is currently available on the recreational fisheries taking place in the vicinity of the DFMPA, it would be useful to include this. The specific characteristics of the fishery(ies) may also pertain to trying to surmise why the reserve effect is measurable for sheephead and whitefish – perhaps recreational fishers target primarily these two species? And getting back to the next steps part of the discussion, the authors could provide more detail on how an expanded sampling design could overcome some of the weaknesses of the original limited research.

One might argue that it would be necessary to conduct this additional research and feed the additional data into the analysis, before the paper could be ready for publication. However, even though the findings are limited and the implications circumscribed, this paper is pioneering and merits publication. With a more robust discussion of what constitutes OECM and how DFMPAs can add value alongside proper MPAs, the paper will provide valuable guidance to MPA planners and to marine spatial planners looking at wider scale marine management and conservation.

One very minor point: in Line 61, recommend substituting ‘very little research’ with ‘very few published studies’ – no one knows how much research is going on, especially if the research is not yielding significant results.

Reviewer #2: In this study, the authors used ROV surveys to compare the density, size structure, and biomass of fishes between a de facto marine protected area (DFMPA) and a fished site on San Clemente Island, California, in the 2-3 years following full closure of the DFMPA. The authors report differences in the density and biomass of one fish species (California sheephead) and biomass of another fish species (ocean whitefish), but no differences in diversity, community structure, or the abundance and biomass of any other species. Below are specific comments for the authors to consider.

- How long has the DFMPA been in place in Zone G? When were the 8 naval safety zones established? What fraction of the time are the other zones closed to human activities, such as fishing? It sounds like the other zones can serve as partial DFMPAs, depending on the frequency with which they are closed due to military activities. OK… I see that in the methods the timing of implementation of the DFMPA was 2010. This should be mentioned in the abstract and probably earlier in the intro as well. If the surveys occurred in 2012 and 2013, then it appears that the area was only closed for 3 years. If that is the case, it’s not surprising that few significant differences were detected, especially if fishing pressure in zone F isn’t too intense.

- Line 133. I think maybe the wrong citation is used for fishing pressure at San Clemente. I think this should be reference 23, not 24. In addition, more detail is needed to quantify how much fishing pressure occurs in zone F, as opposed to more generally at San Clemente Island. Zone F is on the windward side and far from the most popular anchorage, so how much fishing pressure actually occurs there? If not all that much, that could explain why differences between the DFMPA and fished zone were not great for most species.

- One of the major challenges with this study is that it only compares 1 DFMPA to 1 fished area outside the DFMPA. This needs to be discussed in more detail, because the differences or lack of differences observed could be due to this design. The authors include some simple comparisons of habitat between the two areas, suggesting that habitat is similar. However, that analysis used simple metrics of percent rocky habitat and depth. What is not shown is whether the habitat quality is similar or not. How does the rugosity and complexity of the rocky habitat compare between the two zones? If one has mainly flat rock and the other has much more complex 3-dimensional habitat with higher relief, that could explain the differences in sheephead and ocean whitefish density and biomass, irrespective of fishing pressure. The size distribution comparison for sheephead shows that not only are larger fish present in the DFMPA, but also new recruits and small juvenile size classes are present as well. That could be an indicator of better habitat quality for all size classes of this species in the DFMPA. Lastly, Zone G is located at the northern tip of the island. Many studies show higher densities and biomass of fishes at the tips of islands, reefs, and atolls, due to stronger currents, productivity, and prey availability at the tips. This oceanographically-driven phenomenon could be another explanation for higher sheephead density and biomass, that should be discussed.

- More description is needed of the video imagery analysis section. How do you define a non-overlapping video quadrat? What are the dimensions of the quadrat? How many quadrats per transect?

- Even though the community structure was similar in the multivariate analysis, it would still be useful to include the nMDS figure to show the reader how the sites overlap. Where the community structure analyses done using density or biomass and the response variable. Are the results the same for both metrics?

- The analysis comparing size distributions between the DFMPA and fished site should be placed in the results section and not saved for the discussion. Size frequency distribution comparisons should also be shown for some of the other common species sampled.

6. PLOS authors have the option to publish the peer review history of their article (what does this mean?). If published, this will include your full peer review and any attached files.

Reviewer #1: No

Reviewer #2: No

---

## [Author Response · Author response to Decision Letter 0]

26 Dec 2019

Response to reviewers

Reviewer #1

Comment: This is a clear and compelling paper that evaluates some of the effects of restricted use in a de facto marine protected area (DFMPA) off the California coast. The subject is of particular interest as countries look for ways to ramp up the percent protected area coverage of their maritime jurisdictions. Military restrictions around San Clemente Island present an opportunity to study a controlled area and compare biomass and community structure with adjacent fished areas. The paper is technically sound, extremely well-written, and presents the data that were used in the analysis.

Response: We thank the reviewer for these comments.

Comment: However, regarding whether the data can be said to support the conclusions, this is the paper’s greatest weakness. The authors claim that the lack of differentiation in biodiversity metrics between the DFMPA and fished sites for most species surveyed is due to longer trajectories of recovery for species other than sheephead and ocean whitefish – but does what we know about the life histories the other species support this contention? 

Response: In lines 345-347 and 352-353, we have included additional information on the life histories of sheephead and ocean whitefish. In line 346 we note that these life histories differ from some of the other focal species considered.

Comment: An alternative hypothesis might be that fishing pressure in the fished area is spread across many species, and therefore does not alter the community ecology (therefore not presenting a condition from which recovery can be measured). 

Response: We have added this point in lines 395-397.

Comment: Or perhaps microhabitats do vary between fished and DFMPA sites – these were not assessed and as the authors themselves state, may be a bigger factor in determining condition (biomass and diversity) than fishing pressure.

Response: In lines 362-378, we have expanded our discussion of how microhabitat type, distribution, and quality may have influenced our results.

Comment: Differential spillover is unlikely, unless the authors can point to some aspect of reproductive ecology or larval dispersion of the sheephead and whitefish not shared by the other species. 

Response: We did not directly assess differences in reproductive ecology for focal species. We have added some language noting this limitation in lines 386-387.

Comment: Since the data collected and the time since closure are both limited, it may be too early and too much of a stretch to speculate as to why the DFMPA has bigger specimens and higher densities of sheephead and whitefish and not the other species.

Response: We agree. We make this point in the Abstract (lines 38-40) and the first part of the Discussion (lines 326-328).

Comment: I believe the paper would also benefit from an expanded discussion around Other Effective Conservation Measures (OECMs). The authors are undoubtedly aware of the very fluid (and sometimes highly contentious) debates around what constitutes OECMs, and the fact that there is a lot of pressure to make the definition as wide as possible so that contracting parties to the Convention on Biological Diversity can claim to be approaching Aichi Target 11, thereby keeping to their commitments. However, the draft decision of the SBBSTA (CBD/SBSTTA/22/L.2 6 July 2018) does say that the definition of OECM covers considerations of governance, including equity issues, which military bases and restricted areas could not be said to fulfill. Furthermore, while the DFMPA studied does demonstrate tangible conservation benefits for two heavily fished species, no assessment was made of how routine military operations or periodic exercises affect all species in the area, including marine mammals. ‘Effective conservation’ within the OECM means conservation of all species, not only a select few. The paper should be forthright about mentioning these limitations.

Response: We have significantly expanded the portion of the Discussion in which we address OECMs. In lines 429-454, we clarify the connection between the ongoing debate surrounding the definition of an OECM and the need for more consistent, long-term, robust biological and environmental monitoring of DFMPAs. In lines 444-454, we elaborate on several complexities surrounding DFMPAs/OECMs (including issues of equity, access, and potentially harmful human activity) that may not apply to formal MPAs. We then explain how these complexities may complicate the assessment of DFMPA/OECM conservation benefits.

Comment: Another area where the paper could be strengthened is in the discussion of future research – in particular how to overcome the design flaw that the current study has in that it had no reference points in fished areas other than those adjacent to the DFMPA. Spillover could confound the existing comparison, as could any perception issues among fishers that would limit their fishing in the areas adjacent to the restricted area (if, for instance, fishers were wary of accidentally entering the restricted area).

Response: We have significantly expanded the portion of the Discussion that addresses these limitations. In lines 381-393, we elaborate on the issues associated with having our reference site immediately adjacent to the DFMPA. In lines 400-405, we discuss potential future work that could directly address these flaws, including suggesting additional DFMPA and reference sites at San Clemente Island for potential inclusion in a larger-scale sampling design.

Comment: If any information is currently available on the recreational fisheries taking place in the vicinity of the DFMPA, it would be useful to include this. The specific characteristics of the fishery(ies) may also pertain to trying to surmise why the reserve effect is measurable for sheephead and whitefish – perhaps recreational fishers target primarily these two species? 

Response: Although the Navy does not collect data on recreational fishing activity at San Clemente Island, we have obtained relevant recreational fishing data for sheephead and ocean whitefish from the California Department of Fish and Wildlife, and we include these data in lines 338 and 351, respectively.

Comment: And getting back to the next steps part of the discussion, the authors could provide more detail on how an expanded sampling design could overcome some of the weaknesses of the original limited research.

Response: In lines 400-405, we suggest additional DFMPA and reference sites at San Clemente Island for potential inclusion in a future larger-scale sampling design. We also explain how such a design would address some of the weaknesses inherent in this study.

Comment: One might argue that it would be necessary to conduct this additional research and feed the additional data into the analysis, before the paper could be ready for publication. However, even though the findings are limited and the implications circumscribed, this paper is pioneering and merits publication. With a more robust discussion of what constitutes OECM and how DFMPAs can add value alongside proper MPAs, the paper will provide valuable guidance to MPA planners and to marine spatial planners looking at wider scale marine management and conservation.

Response: We thank the reviewer for these comments.

Comment: One very minor point: in Line 61, recommend substituting ‘very little research’ with ‘very few published studies’ – no one knows how much research is going on, especially if the research is not yielding significant results.

Response: We have made the requested revision.

Reviewer #2 

Comment: In this study, the authors used ROV surveys to compare the density, size structure, and biomass of fishes between a de facto marine protected area (DFMPA) and a fished site on San Clemente Island, California, in the 2-3 years following full closure of the DFMPA. The authors report differences in the density and biomass of one fish species (California sheephead) and biomass of another fish species (ocean whitefish), but no differences in diversity, community structure, or the abundance and biomass of any other species. Below are specific comments for the authors to consider.

Response: We thank the reviewer for these comments.

Comment: How long has the DFMPA been in place in Zone G? When were the 8 naval safety zones established? What fraction of the time are the other zones closed to human activities, such as fishing? It sounds like the other zones can serve as partial DFMPAs, depending on the frequency with which they are closed due to military activities. OK… I see that in the methods the timing of implementation of the DFMPA was 2010. This should be mentioned in the abstract and probably earlier in the intro as well. 

Response: We have added the closure date to the Abstract (line 28) and the Introduction (line 99). 

Comment: If the surveys occurred in 2012 and 2013, then it appears that the area was only closed for 3 years. If that is the case, it’s not surprising that few significant differences were detected, especially if fishing pressure in zone F isn’t too intense.

Response: We agree. We make this point in the Abstract (lines 38-40) and the Discussion (lines 326-328).

Comment: Line 133. I think maybe the wrong citation is used for fishing pressure at San Clemente. I think this should be reference 23, not 24.

Response: We agree. We have replaced reference 24 with reference 23 in line 133.

Comment: In addition, more detail is needed to quantify how much fishing pressure occurs in zone F, as opposed to more generally at San Clemente Island. Zone F is on the windward side and far from the most popular anchorage, so how much fishing pressure actually occurs there? If not all that much, that could explain why differences between the DFMPA and fished zone were not great for most species.

Response: We have added this point in lines 393-397.

Comment: One of the major challenges with this study is that it only compares 1 DFMPA to 1 fished area outside the DFMPA. This needs to be discussed in more detail, because the differences or lack of differences observed could be due to this design. 

We have expanded the section of the Discussion that addresses this issue. In particular, in lines 400-405, we suggest additional DFMPA and reference sites at San Clemente Island for potential inclusion in a future larger-scale sampling design. We also explain how such a design would address some of the weaknesses inherent in this study.

Comment: The authors include some simple comparisons of habitat between the two areas, suggesting that habitat is similar. However, that analysis used simple metrics of percent rocky habitat and depth. What is not shown is whether the habitat quality is similar or not. How does the rugosity and complexity of the rocky habitat compare between the two zones? If one has mainly flat rock and the other has much more complex 3-dimensional habitat with higher relief, that could explain the differences in sheephead and ocean whitefish density and biomass, irrespective of fishing pressure.

Response: We have added this point in lines 377-378.

Comment: The size distribution comparison for sheephead shows that not only are larger fish present in the DFMPA, but also new recruits and small juvenile size classes are present as well. That could be an indicator of better habitat quality for all size classes of this species in the DFMPA. 

Response: We have added this point in lines 368-371.

Comment: Lastly, Zone G is located at the northern tip of the island. Many studies show higher densities and biomass of fishes at the tips of islands, reefs, and atolls, due to stronger currents, productivity, and prey availability at the tips. This oceanographically-driven phenomenon could be another explanation for higher sheephead density and biomass, that should be discussed.

Response: We have added this point in lines 397-400.

Comment: More description is needed of the video imagery analysis section. How do you define a non-overlapping video quadrat? What are the dimensions of the quadrat? How many quadrats per transect?

Response: We have revised the imagery analysis section in lines 179-184 to more clearly describe how ecological and habitat data were extracted from underwater imagery. We have deleted the term “video quadrat.”

Comment: Even though the community structure was similar in the multivariate analysis, it would still be useful to include the nMDS figure to show the reader how the sites overlap. Where the community structure analyses done using density or biomass and the response variable. Are the results the same for both metrics?

Response: We have included the NMDS plot as a new figure (Fig 5). This analysis was based on Bray-Curtis dissimilarity indices between fished and DFMPA transects, which is calculated using counts of unique species per area surveyed.

Comment: The analysis comparing size distributions between the DFMPA and fished site should be placed in the results section and not saved for the discussion. Size frequency distribution comparisons should also be shown for some of the other common species sampled.

Response: We have moved the discussion of this analysis to the Results section (lines 299-302). We have updated Fig 4 to include size frequency distribution comparison for Ocean Whitefish, the only other focal species for which biomass was significantly higher at the DFMPA site.

 

Academic Editor

Comment: Overall, a really interesting and well written manuscript. The authors introduce a lot of abbreviations, many of which are used again only once. The PLoS formatting rule is that they should be used three times, otherwise just spell them out.

Response: We thank the academic editor for these comments. We have spelled out abbreviations for any term used less than three times in the manuscript.

Comment:

Line 73: reference not cited corrected. Should be ‘Roberts et al. [ref #]’

Line 76: Don’t capitalise ‘refuge’

Line 77: reference not cited corrected. Should be ‘Rogers-Bennett et al. [ref #]’

Line 96: ‘scuba’ is no longer considered an acronym, but just a regular word, so do not write is all capitals

Line 117: The comma should be a semicolon

Line 118: The comma should be a semicolon

Line 119: The comma should be a semicolon

Line 127: Write as ‘at two sites in the northwest corner’

Line 129: Should spell out ‘SEAL’ here

Line 161: The comma should be a semicolon

Line 162: The comma should be a semicolon

Line 169: The dash should be a comma

Line 179: The abbreviation LWR is not used in the manuscript again so please delete

Line 182: Supporting information is written as ‘S1 Table’ so please correct throughout manuscript and in the supporting information itself.

Line 202: S2 Table

Line 204: The abbreviation FAMD is only used once more so do not introduce it here and just spell it out in the rest of the manuscript.

 Also, do not capitalise ‘factor analysis of mixed data’

Line 206: Do not capitalise ‘principal component analysis’

Line 207: Do not capitalise ‘multiple correspondence analysis’

Line 220: The abbreviation GLM is used in several tables but it not defined in the text so please introduce the abbreviation here

Line 225: The semicolon should be a comma

Line 254: S3-4 Tables

Line 276: While AIC has been defined in the text, AICc and AICw have not, so there should be a note for the table explaining what they mean (and for all the tables that use them).

Line 279: Same as above

Line 282: Same as above

Line 296: Same as above

Line 300: delete the duplicate ‘hint at’

Line 310: Replace the dash with a comma and say ‘a ten-fold’

Line 313: I would say ‘fishers’ rather than ‘fishermen’

Line 313: replace the semicolon with a full stop and start the next sentence ‘Since large males….’

Lines 325-326: This is a bit strange and out of place because this is not mentioned in the methods or results and I am not sure why. The authors should mention this comparison was made in the methods and report the result in the results section, rather than here.

Line 368: Do not introduce the abbreviation ‘CBD’ as it is only used once more

Line 369: What is the reference for this quote?

Line 375: Do not introduce the abbreviation ‘OECMs’ as it is only used once more

Line 376: Spell out CBD

Line 377: Spell out OECM

Response: We have made the requested revisions.

---

## [Editor Report · Decision Letter 1]

2 Jan 2020

Early conservation benefits of a de facto marine protected area at San Clemente Island, California

PONE-D-19-27751R1

Dear Dr. Esgro,

We are pleased to inform you that your manuscript has been judged scientifically suitable for publication and will be formally accepted for publication once it complies with all outstanding technical requirements.

With kind regards,

Heather M. Patterson, Ph.D.

Academic Editor

PLOS ONE
---

## [Editor Report · Acceptance letter]

8 Jan 2020

PONE-D-19-27751R1 

Early conservation benefits of a de facto marine protected area at San Clemente Island, California 

Dear Dr. Esgro:

I am pleased to inform you that your manuscript has been deemed suitable for publication in PLOS ONE. Congratulations! Your manuscript is now with our production department. 

With kind regards,

on behalf of

Dr. Heather M. Patterson 

Academic Editor

PLOS ONE